Analysis of pedestrian activity before and during COVID-19 lockdown, using webcam time-lapse from Cracow and machine learning

http://orcid.org/0000-0001-8161-9929 Szczepanek Robert robert@szczepanek.pl
Faculty of Environmental and Power Engineering, Cracow University of Technology , Cracow , Poland
Benfield Mark
Electronic publication date: 2020 Oct 5
Publication date: 2020
Volume: 8
Electronic Location ID: e10132
Received 2020 May 12; Accepted 2020 Sep 18
Copyright: © 2020 Szczepanek
Copyright year: 2020
Copyright holder: Szczepanek
License: This is an open access article distributed under the terms of the Creative Commons Attribution License, which permits unrestricted use, distribution, reproduction and adaptation in any medium and for any purpose provided that it is properly attributed. For attribution, the original author(s), title, publication source (PeerJ) and either DOI or URL of the article must be cited.
License URL: https://creativecommons.org/licenses/by/4.0/

Keywords: YOLOv3, COVID-19, OpenCV, Webcam, Pedestrian counting, People detection, Cracow, Data science, Database

Funding: The author received no funding for this work.

==============================
At the turn of February and March 2020, COVID-19 pandemic reached Europe. Many countries, including Poland imposed lockdown as a method of securing social distance between potentially infected. Stay-at-home orders and movement control within public space not only affected the touristm industry, but also the everyday life of the inhabitants. The hourly time-lapse from four HD webcams in Cracow (Poland) are used in this study to estimate how pedestrian activity changed during COVID-19 lockdown. The collected data covers the period from 9 June 2016 to 19 April 2020 and comes from various urban zones. One zone is tourist, one is residential and two are mixed. In the first stage of the analysis, a state-of-the-art machine learning algorithm (YOLOv3) is used to detect people. Additionally, a non-standard application of the YOLO method is proposed, oriented to the images from HD webcams. This approach (YOLOtiled) is less prone to pedestrian detection errors with the only drawback being the longer computation time. Splitting the HD image into smaller tiles increases the number of detected pedestrians by over 50%. In the second stage, the analysis of pedestrian activity before and during the COVID-19 lockdown is conducted for hourly, daily and weekly averages. Depending on the type of urban zone, the number of pedestrians decreased from 33% in residential zones to 85% in tourist zones located in the Old Town. The presented method allows for more efficient detection and counting of pedestrians from HD time-lapse webcam images compared to SSD, YOLOv3 and Faster R-CNN. The result of the research is a published database with the detected number of pedestrians from the four-year observation period for four locations in Cracow.

Introduction

The COVID-19 pandemic that appeared in Europe in early 2020 has a major impact on societies around the world. Its economic, social and environmental impact affect many citizens. Many countries have introduced extraordinary restrictions related to transport and use of public spaces. The direct consequence of this situation is a significant decrease in the number of pedestrians in public space.

Cracow is one of the most popular tourist cities in Poland (Central Europe). It is also an academic center with the oldest university in Poland, the Jagiellonian University founded in 1364 by Casimir the Great. With 771,069 inhabitants in 2018 and a population density of 2,359 person/km2 (Rozkrut, 2019), Cracow is the second largest city in Poland. Being one of the oldest cities with many tourist attractions, virtually all year round the center of the Old Town is visited by many tourists from home and abroad.

In Poland, the first case of COVID-19 was officially confirmed on 4 March 2020. On 13 March 2020, the Polish government announced the first restrictions related to COVID-19. This included limiting the activities of shopping centers, restaurants, bars and cafes, closing swimming pools and gyms. A significant reduction in mobility was introduced on March 24 (Jarynowski et al., 2020). The ban on leaving home did not only include going to work, a store or a pharmacy. Additional restrictions on the operation of markets were introduced on March 31.

Patterns of human activity in the urban environment depend on several factors such as, for example, night lighting (Wang et al., 2019), but the impact of formal restrictions on movement in public space is rarely considered and analyzed. In addition to public video surveillance systems, there are also private video monitoring systems that can also be used to detect and count people. Some of them have been operating continuously for several years, enabling comparative studies with pre-pandemic periods.

The study has two main goals: to evaluate the YOLOv3 people detection algorithm on images from HD webcams and application of YOLOv3 to assess changes in pedestrian activity in public space before and during COVID-19 lockdown in Cracow, based on the hourly webcam time-lapse.

Social distance during COVID-19

Wellenius et al. (2020) used anonymous and aggregated mobility data (Aktay et al., 2020) to assess social distance in the United States during COVID-19. The impact of the social distance order was very different in each state, from a 36% drop in displacement New Jersey to a 12% drop in Louisiana. The most effective ban was to impose restrictions on the work of bars and restaurants, which was associated with a 25.8% reduction in people’s activity. Wellenius et al. (2020) concludes that public procurement seems to be very effective in encouraging people to stay at home to minimize the risk of COVID-19 transmission. In the case of Poland, in the COVID-19 Community Mobility Report (29 March 2020), mobility trends in places such as restaurants, cafes, shopping centers, theme parks and museums fell by 78%. In the case of the Lesser Poland Voivodship in which Cracow is located, this decrease is 84% (Aktay et al., 2020). Social behavior has a fundamental impact on the dynamics of the spread of infectious diseases (Prem, Cook & Jit, 2017). Inhabitants of larger Polish cities are more afraid of overcrowded hospitals and inefficient healthcare than small towns and villages (Jarynowski et al., 2020).

The Center for Science and Systems Engineering (CSSE) at Johns Hopkins University provides daily data updates via COVID-19 Data Repository (Dong, Du & Gardner, 2020). The first confirmed cases of COVID-19 in Italy and Spain were identified at the end of February 2020 (Saglietto et al., 2020). The lockdown has been widely used in Italy since March 8 and in Spain since March 16. Restrictions on citizens’ mobility have reduced disease transmission in both countries (Tobías, 2020).

Chinazzi et al. (2020) findings indicate that 90% of travel restrictions to and from mainland China only modestly affect the epidemic trajectory unless combined with a 50% or higher reduction of transmission in the community. Fang et al. (2020) uses the crowd flow model for virus transmission to simulate the spread of the virus caused by close contact during pedestrian traffic. Mobility restrictions are important (Arenas et al., 2020; Ferguson et al., 2020) or sometimes crucial (Mitjà et al., 2020), but as shown by Mello (2020) the number of people crossing each other can be drastically reduced if one-way traffic is enforced and runners are separated from walkers. To properly quantify the transmission of an epidemic, the spatial distribution of potential disease hazards (e.g., crowd) should be assessed (Ng & Wen, 2019; Fang et al., 2020). Webcams can be a potential source of such information.

People detection

Object detection is one of the rapidly growing areas of computer vision. Proper detection of people is crucial for autonomous cars, advertising planning and many other industries and public safety. Kajabad & Ivanov (2019) proposed a method of finding areas more attractive to customers (hot zones) based on people detection. Sometimes, people must be detected in a heavy industry environment (Zengeler et al., 2019) or in hazy weather (Li, Yang & Qu, 2019). A lot of research is being done to detect objects in a variety of environments, but this is not just about detecting people. Computer vision methods are used to count species in environmental research: 1 minute time-lapse for fish passage and abundance in streams (Deacy et al., 2016), or 5 min time-lapse for bears counting (Deacy et al., 2019). There are two main approaches to detecting a person or other object in the image. The first approach is based on computer vision techniques, the second on deep learning algorithms. Comprehensive survey on computer vision and deep learning techniques for pedestrian detection and tracking is presented by Brunetti et al. (2018).

Computer vision

Traditional pedestrian detectors have been known for over two decades. They are based on the representation of the features of objects obtained from computer vision. Oren et al. (1997) proposed the use of Haara waves in 1997, and Maliniowski in 2005 the use of the Oriented Gradient (HOG) Histogram. Also Local Binary Patterns (LBP) (Ojala, Pietikainen & Maenpaa, 2002) can be used for pedestrian detection (Zheng et al., 2010). Among them, HOG and its variations are considered the most successful hand-engineered features for pedestrian detection (Liu, Copin & Stathaki, 2016a; Liu et al., 2019b). For visual surveillance applications, background subtraction method can also be used (Maddalena & Petrosino, 2008). In hybrid implementation of computer vision methods, pedestrian detection on the basis of 2D/3D LiDAR data and visible images of the same scene are applied (Hasfura, 2016; El Ansari, Lahmyed & Trémeau, 2018).

Deep learning

In recent years, several convolutional neural networks (CNN) models for object detection have been proposed (Ren et al., 2018): R-CNN in 2014; Fast R-CNN in 2015 and Faster R-CNN in 2015. These two-step detection algorithms divide the problem into two stages: (i) generating region proposals and (ii) classification of candidate regions. But these traditional deep learning algorithms suffer from low speed (Kajabad & Ivanov, 2019). To overcome this limitation, Redmon et al. (2016) proposed a one-step detection algorithm called YOLO (You Only Look Once), enabling easy implementation end-to-end object detection. Further improvements of this algorithm are known as YOLO9000 (or YOLOv2) (Redmon & Farhadi, 2017) and YOLOv3 (Redmon & Farhadi, 2018). Second popular one-stage algorithms is RetinaNet (Lin et al., 2017). It deals with the problem of the extreme foreground-background class imbalance encountered during the training of dense detectors and proposes a new solution to this problem. The third detection algorithm is the Single Shot MultiBox Detector (SSD). The core of SSD is predicting category scores and box offsets for a fixed set of default bounding boxes using small convolutional filters applied to feature maps (Liu et al., 2016b). To improve model performance for small objects, SSD applies additional data augmentation strategy. All three algorithms achieve state-of-the-art speed and accuracy (Zengeler et al., 2019), so they can be used in real-time applications. The CNN-based approaches provide significant improvements over traditional approaches across all datasets (Sindagi & Patel, 2018).

The YOLO model applies a single neural network to the complete image. It looks at the whole image at test time so its predictions are based on the global context in the image. This network divides the image into regions and predicts bounding boxes and probabilities for each region. These bounding boxes are weighted by the predicted probabilities. YOLOv3 predicts an objectness score for each bounding box using logistic regression (Redmon & Farhadi, 2018). During training, the binary cross-entropy loss is used for class predictions. The original YOLO model trains the classifier network at 224 × 224 and increases the resolution to 448 × 448 for detection (Redmon & Farhadi, 2017). Backbone for YOLOv3 is Darknet-53 network, and standard image sizes are 320 × 320. The Darknet-53 network is composed of 53 consecutive 3 × 3 and 1 × 1 convolutional layers. YOLOv3 makes detection at three different scales downsampling the dimensions of the input image by 32, 16 and 8. Darknet architecture is a pre-trained model for classifying 80 different classes. Several improvements to the YOLO model have been proposed for detecting people (Putra et al., 2017, 2018; Lan et al., 2018; He et al., 2019; Li, Yang & Qu, 2019), but even standard YOLOv3 outperforms traditional computer vision methods and most of deep neural network methods (Ghosh & Das, 2019; Zengeler et al., 2019; Kajabad & Ivanov, 2019; Yun et al., 2018). YOLOv3 achieves 93.8% top-five score on the COCO dataset (Redmon & Farhadi, 2018).

Pre-trained networks for standard image sizes are available in several repositories, enabling fast and relatively easy application of YOLO model. The squeeze YOLO-based people counting (S-YOLO-PC) proposed by Ren et al. (2018) can detect and count people with 41 frames per second (FPS) with the Average Precision (AP) of 72%. Feng, Lin & Lin (2019) reports YOLOv2 mean Average Precision (mAP) of 76.8%, which is very close to 78.6% reported by authors of YOLO method (Redmon & Farhadi, 2017). However, even the latest version of YOLOv3 has some limitations. If there are two anchor boxes but three objects in the same grid cell, it does not support them correctly, which ultimately leads to missing objects (Kajabad & Ivanov, 2019). YOLO achieves about 10% missing detection rate for pedestrian detection (Lan et al., 2018). Yun et al. (2018) reports that YOLOv3 default architecture achieves the mAP of 42.7%.

Pedestrian detection

Scale problem

Robustly detecting pedestrians with a large variant on sizes and with occlusions remains a challenging problem (Liu et al., 2019a, 2019b). Pedestrian detection is limited by image resolution and complexity of the background scene. Effective detector should be able to detect people at different scales. Liu, Elmikaty & Stathaki (2018) present a method where a large-size pedestrian should be represented by features from deep layers, whereas a small-size pedestrian should be represented by features from shallow layers which are of higher resolutions. Liu et al. (2019a) proposed a gated feature extraction framework consisting of squeeze units, gate units and a concatenation layer which perform feature dimension squeezing, feature elements manipulation and convolutional features combination from multiple CNN layers. The Faster R-CNN is also used as benchmark for detecting occluded pedestrians with results comparable to fine tuned models (Liu et al., 2019b; Zhang, Yang & Schiele, 2018). However, Lin et al. (2020) and Zhang et al. (2016) state that convolutional feature maps of the this classifier are of low resolution for detecting small objects. Evaluation of average precision and the tuning of the model is usually limited to objects in the 50–100 m range, as in the CityScapes Dataset for Semantic Urban Scene Understanding (Cordts et al., 2016). In research by Dollar et al. (2011), pedestrians represented by 30 pixels or less are treated as distant objects. The main source of false negative classification according to Zhang et al. (2017) is mainly the small scale, therefore the authors only consider pedestrians with a height of more than 30 pixels. Scale-oriented models such as Scale-Aware Fast R-CNN are developed (Li et al., 2017), but even in this case, small objects are about 50 pixels high.

Crowd counting

The next issue in urban space or during mass events is the crowd. There are mainly three types of methods to count the number of people in the crowd from video (Ren et al., 2018): (i) statistical method to estimate the number of people in a region, (ii) combination of object detection with object tracking and (iii) use of path information of the points, with subsequent cluster analysis of the feature point path. People detection in crowded spaces is the most challenging task, because of the people occlusions (Stewart, Andriluka & Ng, 2016; Kajabad & Ivanov, 2019). Crowd counting requires development of new methods (Stewart, Andriluka & Ng, 2016; Lei et al., 2020) like Dynamic Region Division (He et al., 2019). Yang et al. (2020) proposes counting crowds using a scale-distribution-aware network and adaptive human-shaped kernel. Existing crowd counting methods require object location-level annotation or weaker annotations that only know the total count of objects (Lei et al., 2020). Cheng et al. (2020) proposed an FFPM model for the pedestrian detection using body parts (head, shoulders, hands, knees and feet) followed by full body boosting model and a classification layer. This approach works well in a crowded environment with partially obscured pedestrians. Model proposed by Jiang et al. (2019) combines the classic computer vision approach (HOG+LBP features) with the GA-XGBoost deep learning algorithm.

Materials and Methods

Study area

Images from webcams are collected in Cracow, the second largest city in the country and the capital of the Lesser Poland Voivodeship. Cracow is divided into the medieval Old Town, located in the center and the surrounding residential and industrial zones. The Vistula, the largest river in Poland, flows through the city center.

Two of the webcams are located on the Royal Road, going from Wawel Castle through Main Square to north of the city. These webcams are named All Saints’ Square and Grodzka (Fig. 1). Grodzka Street has a tourist character, and All Saints’ Square, being in the tourist zone as shown in Table 1, is also an important communication point in the city. The third webcam (Wawel Castle) is located in the tourist/residential zone. Parking for tourists visiting Wawel Royal Castle is adjacent to the riverside promenade, which is used by residents. The fourth webcam (Podgorze Market Square) is a typical residential zone located on the other side of the Vistula river (Fig. 1).

Figure 1 Location in Cracow (Poland) and approximate field of view for webcams used in these studies (www.webcamera.pl).

Technical details in Table 1. Map data from OpenStreetMap (https://www.openstreetmap.org/).

Table 1 Webcam visibility range and source image URLs.

Pedestrians area refers to the part of the area accessible to pedestrians.

Webcam name	Distance to pedestrians (m)	Pedestrians area (ha)	Urban zone/URL	
Wawel Castle	50–400	0.92	Touristic/residential	
			https://krakow2.webcamera.pl/	
All Saints’ Square	10–150	0.32	Touristic mainly	
			https://krakow1.webcamera.pl/	
Grodzka	10–100	0.14	Touristic	
			https://hotel-senacki-krakow.webcamera.pl/	
Podgorze Market Square	30–120	0.31	Residential	
			https://krakow3.webcamera.pl/	

Due to the medieval nature of the area, cameras from Royal Road have a very narrow field of view. The webcam on All Saints’ Square is located on a small square, so in fact most of the visible pedestrian area belongs to Grodzka Street. This webcam is also in the lowest position among all four, enabling easier detection of pedestrians due to the short distance from the detected objects. The Wawel Castle webcam with probably the most beautiful view from all Cracow webcams has the largest distance to detected pedestrians and is the highest mounted webcam (the sixth floor).

All webcams, shown in Table 1, are publicly available and broadcast live via dedicated websites, but access to the ad-free version is limited due to the commercial nature of the service. Webcamera.pl is probably one of the largest providers of streaming cameras in Poland, with a long history and almost 350 webcams located all over Poland. To make detection results comparable, webcams with a moving field of view were excluded from the analysis, although they are in very good locations, such as the Main Square, the largest medieval town square in Europe (https://krakow4.webcamera.pl/).

Webcam time-lapse

Webcam time-lapse is made and downloaded every hour (Fig. 2), directly from www.webcamera.pl provider. In this study, approximately 33,800 images were collected and used for each webcam from 9 June 2016 to 19 April 2020. The analysis is based on 1,412 days (201 weeks) of continuous observation. The total size of the set of hourly time-lapse images for four webcams in this period exceeds 10 GB.

Figure 2 Sample image from the All Saints’ Square webcam at midnight, with pedestrians and cars detected by YOLO.

Timestamp: 29 October 2016 00:00. Photo credit: www.webcamera.pl.

Methods

In the first part of these studies, only the problem of the size of the object (person) is considered. Due to the convenient location of the webcams (between the first and sixth floors) and low to moderate density of pedestrians, crowd counting methods can be omitted.

The standard Darknet-53 architecture with the YOLOv3 model is used as the main pedestrian detector. The interface to the model is built in the Python 3 programming language, in the main script yolo_count.py. All code is available in the public repository (https://gitlab.com/Cracert/pedestrian-count-covid-19) under the MIT license. The OpenCV library (Bradski, 2000) with built-in support for the Darknet architecture is used in these studies as a machine learning platform. From the pre-trained Darknet architecture, only the first 9 classes (from 80) are saved during calculations. These classes (person, bicycle, car, motorcycle, airplane, bus, train, truck, boat) are directly related to the urban space and can be used for other research. The results of pedestrian detection are counted and saved in CSV files for each year of the webcam, with one row corresponding to 1 h.

The pre-trained YOLOv3 model weights for people detection are available for direct use, so the training phase can be omitted (https://pjreddie.com/media/files/yolov3.weights). One of the standard image resolutions for training is 416 × 416, while the source HD webcam image resolution used in this study is 1,280 × 720. As a first approach, YOLOv3 is applied directly to the collected images (Fig. 2). The second approach assumes that the right image ratio can improve the average precision of the model. The input images are divided into 6 almost square tiles 426 × 360 (Fig. 3). This reduces the image ratio from 1.78 (Fig. 2) to 1.18 (Fig. 3) and makes the image’s proportions more similar to the training data set. In addition, not all tiles contain pedestrian areas, so some tiles can be omitted in the calculation, which significantly reduces detection time.

Figure 3 The upper left two tiles from the split image (Fig. 2).

Pedestrians (A) undetected and (B) detected by YOLOtiled method. Photo credit: www.webcamera.pl.

The workflow in the YOLOtiled model can be described with the following steps:Split the source image into square tiles of a size similar to the image size when training the model.

For further calculations, only select the tiles in the potential pedestrian zone. Tiles containing only buildings or sky can be excluded.

Run the YOLO model on each selected tile and aggregate the results. In fact, this method can be applied to any model, not just YOLO.

Number of detected pedestrians is saved in data folder as CSV files. Each file contains a header with the main detected classes and data containing a timestamp (day and hour) with the corresponding number of detected objects. One row corresponds to one hour time-lapse. Further analysis and visualization takes place in Jupyter notebooks using the pandas library. For the purposes of this article, the YOLOv3 method will be named YOLO from this place, and the method of splitting one HD webcam image into six tiles will be named YOLOtiled.

Model performance verification is based on two additional, state-of-the-art deep learning models (SSD, Faster R-CNN), implemented using the GluonCV framework (Guo et al., 2020). Pre-trained Resnet50 VOC network was used in both models. Ground truth data was prepared by manually counting pedestrians on webcam hourly snapshots in March 2020 (almost 3,000 images in total). The calculation were made on an AMD Ryzen 5 2600X, 16 GB RAM, running 64 bit GNU/Linux Mint 19.1 system, with the use of CPU only, without use of the GPU. Model performance is assessed by evaluating Mean Absolute Error (MAE) and Root Mean Squared Error (RMSE): (1) MAE=1n∑i=0n−1|yi−y^i|

(2) RMSE=1n∑i=0n−1(yi−y^i)2

In both formulas, n is the number of observations (images), yi is the actual number of pedestrians, and y^i is the predicted number of pedestrians from one webcam image.

The comparison of the YOLO and YOLOtiled methods is based on statistical analysis. The number of pedestrians detected from each time-lapse (hour) enables the identification of extreme and mean differences between the two methods. Cases of extreme differences are examined manually to find problems associated with each method. In addition, the sum of detected pedestrians for the webcam over the entire period is used to detect the overall relative difference between YOLO and YOLOtiled. In the second part of the study, a better method was used to assess the change in pedestrian numbers before and during COVID-19.

The image data provider returns the last image, so if the webcam fails, the same last recorded image is returned, resulting in a constant number of pedestrians over time. By analyzing such anomalies, you can determine the dates of webcam malfunction. The verification of the source image data based on the pedestrian number change analysis can be replicated in the supplied Jupyter notebook (analysis-pedestrians.ipynb). Doubtful periods are excluded from further analysis.

The webcam observation time is divided into (i) before the COVID-19 period, 9 June 2016– 13 March 2020 (1,374 days/196 weeks) and (ii) during the COVID-19 period, 13 March 2020–19 April 2020 (38 days/5 weeks). The number of detected pedestrians for these two periods is aggregated into days and weeks using mean values. This makes it easier to visualize trends and generalize results. The mean number of pedestrians from the hourly period before and during COVID-19 is used for the final evaluation. A change in this value corresponds to changes in pedestrian activity over time. It is assumed that hourly snapshots (time-lapse) from webcams are representative for evaluation of relative change in pedestrian activity. However, the method presented is not suitable for determining the absolute number of pedestrians traveling through the analyzed area.

Results

The overall results of pedestrian detection performance by the three deep learning models and the proposed YOLOtiled model are presented in Table 2. The performance of the YOLO model compared to the leading Faster R-CNN is 2% worse taking into account Mean Absolute Error and 10% in terms of Root Mean Squared Error. The YOLOtiled model performance is superior to all state-of-the-art pedestrian detection models. Compared to the top Faster R-CNN model, the improvement is 20% for MAE and 13% for the RMSE. Compared to the original YOLO model, the improvement is approximately 22% for both MAE and RMSE. Considering the image processing time, YOLO is twice as fast as the second fastest model (SSD). Dividing the HD webcam image into six tiles for YOLOtiled model for four cameras in Cracow resulted in 4.4 times longer processing time. However, it is still 2.5 times faster compared to Faster R-CNN.

Table 2 Model performance in pedestrian detection with average processing time per image.

Image processing time includes reading the file from disc and model prediction.

Model	MAE	RMSE	Time (s)	
SSD	9.87	14.32	1.46	
Faster R—CNN	5.38	9.16	8.35	
YOLO	5.48	10.23	0.75	
YOLOtiled (proposed)	4.28	7.96	3.28	

The first part of the research focuses on assessing the YOLO pedestrian detection method and comparing with YOLOtiled. The number of detected pedestrians (people) for each hourly image from four webcams in Cracow from 2016 to 2020 is saved for YOLO and YOLOtiled method.

On average, YOLO results are underestimated compared to the YOLOtiled method. Webcams located at Royal Road (All Saints Square and Grodzka) had the highest absolute detected pedestrian differences up to 50 person, as shown in Table 2. The other two webcams, located in the residential and mixed zone, had differences of less than 25 people.

The mean number of detected pedestrians per image by YOLOtiled model depends on the type of urban zone, with 16.6 pedestrians in the tourist zone and 0.9 pedestrians in the residential zone. The mean difference of detected pedestrians between the two methods is the same, with values exceeding 4.4 in the tourist zone and below 0.4 in the residential zone.

Opposite cases are also reported when YOLOtiled detects fewer pedestrians, but in this case the absolute difference does not exceed 12 pedestrians, as shown in Table 3. The maximum number of detected pedestrians also corresponds to the location of the webcam. Tourist locations in the Old Town (All Saints Square and Grodzka) record up to 80 pedestrians in one image (Fig. 4), and in residential zones below 35. Simply cutting one large image into six smaller tiles significantly increases the number of correctly detected pedestrians (Fig. 4). The detection range also increases, but a certain pedestrian detection threshold is clearly visible at the horizontal cutting height of the tiles.

Table 3 Statistics of detected pedestrian number by YOLO and YOLOtiled method.

	Webcam	
	Wawel Castle	All Saints’ Square	Grodzka	Podgorze Market Square	
max(YOLOtiled − YOLO)	15	50	49	24	
mean(YOLOtiled − YOLO)	0.36	5.70	4.43	0.33	
max(YOLO − YOLOtiled)	12	12	5	7	
max(YOLOtiled)	16	80	58	34	
mean(YOLOtiled)	0.5	16.6	7.0	0.9	
sum(YOLO)	4,064	369,497	86,538	17,642	
sum(YOLOtiled)	16,367	562,122	236,252	28,749	
Detection difference (%)	+302	+52	+173	+63	

Figure 4 Pedestrian detection from the All Saints’ Square webcam by the (A) YOLO and the (B) YOLOtiled method.

Both views are framed to the central part. YOLOtiled view without the upper left tile. Photo credit: www.webcamera.pl.

The total sum of detected pedestrians over the entire period (almost 4 years) is from about 4,000 for YOLO from Wawel Castle webcam to over 500,000 for YOLOtiled from All Saints Square. The relative detection differences between YOLO and YOLOtiled are significant and range from 52% on All Saints Square to 302% at Wawel Castle. This difference is proportional to the mean distance from pedestrians, as shown in Table 1.

Over long distances YOLOtiled can detect significantly more pedestrian than YOLO. An example of such a case is shown on results from All Saints Square (Fig. 4) and from Wawel Castle webcam (Fig. 5). Wawel Castle webcam has the longest distance from pedestrians, from about 50 m to about 400 m. Also in this case, the detection range of pedestrians does not exceed about 200 m. Pedestrians in Fig. 5C, near the detected boat at the upper part, are also not recognized.

Figure 5 The biggest difference from the Wawel webcam, with more pedestrians detected by the YOLOtiled method.

Results from (A) YOLO method, and (B) and (C) two bottom left tiles from the YOLOtiled method. Photo credit: www.webcamera.pl.

The histogram of the difference in pedestrian detection YOLOtiled–YOLO is asymmetrical (Fig. 6). This also applies to other webcams. The difference of zero is dominant for all webcams, but mean value of 5.70 for the All Saints Square camera, as shown in Table 3, compared to the extreme number of detected pedestrians on one image in the range of 50–80 makes this difference significant. As a result, the YOLOtiled method is selected as a better representation of the actual number of pedestrians on webcam time-lapse. With the awareness that this value is also underestimated in relation to the actual number of pedestrians on one image. Assuming that the detection range for both methods is constant (YOLO and YOLOtiled), this should not significantly affect the estimation of the relative change in the number of pedestrians.

Figure 6 The difference in the number of pedestrians detected (YOLOtiled–YOLO) for All Saints’ Square webcam.

The YOLOtiled method, which is a better detector, is used as the basis for the second part of research related to estimating pedestrian activity before and during COVID-19. Data analysis enabled the identification of periods during which camera malfunction or camera data transmission was highly likely. These periods were removed from the dataframe and treated as no data. A detailed analysis with relevant comments can be found in analysis-pedestrians.ipynb Jupyter notebook. A few short periods have been removed from the dataframe for All Saints Square and Grodzka webcams (Fig. 7A). Another problem was identified in Podgorze Market Square, where two periods are characterized by significantly different average values of detected pedestrians. It was found that in mid-2019 the horizontal angle of the webcam was changed, which changed the field of view. In order to maintain the possibility of comparison with the current period (COVID-19), it was decided to abandon the first part of the dataframe (Fig. 7B).

Figure 7 Mean weekly number of pedestrians from hourly time-lapse for (A) two tourist locations and (B) two residential (mixed) locations.

Logarithmic scale on both plots for better visualization of annual cycles in tourist locations.

The high temporal variability of hourly data makes it difficult to visualize the result. For this reason, daily and weekly data aggregation is used for visual analysis.

Figure 7 contains plots of weekly averages for two types of zones in Cracow. The seasonal cycle in the tourist zone is associated with the summer season, while in a residential zone this seasonal cycle is not visible. In the tourist zone, the mean number of pedestrians does not fall below one person per image (logarithmic scale in Fig. 7), while in the residential zone the level is lower by an order of magnitude. There are no visible trends in the number of pedestrians during these four years, but the COVID-19 lockdown is clearly visible in the last weeks of the analyzed period in Fig. 7. The quantitative analysis of this change is presented in the Table 4. Data from the Wawel Castel webcam are more difficult to interpret due to the large distance from pedestrians, which results in a very low detection rate. Therefore, data from this webcam is underestimated and caution should be exercised. Also the results from Podgorze Market Square are difficult to interpret due to the relatively short period of homogeneous observations.

Table 4 Mean number of pedestrians detected by YOLOtiled method from hourly time-lapse, before and during COVID-19.

	Webcam	
	Wawel Castle	All Saints’ Square	Grodzka	Podgorze Market Square	
Before COVID-19	0.49	16.86	7.13	1.66	
During COVID-19	0.14	2.04	0.57	0.82	
Change (%)	−54.64	−78.41	−85.32	−33.82	

Before COVID-19, the All Saint Square webcam registered about ten times as many pedestrians compared to Podgorze Market Square (Table 4). During COVID-19 this ratio changed to 2:1. The largest decrease in the number of pedestrians (85%) is observed on the Grodzka camera, which is a typical tourist destination, and the lowest on Podgorze Market Square (34%) in the residential zone. Mixed urban zones, with tourist and residential activities, report a moderate decrease in pedestrian numbers, from 55% to 78%. About 1,000 hourly time-lapse images during COVID-19 and 33,000 images before this period for each webcam is a long enough time series to draw final conclusions.

Discussion

Performance of state-of-the-art models on ground truth data is consistent with the findings of other authors. The Faster R-CNN model offers the smallest errors, while the YOLO is the fastest. The proposed YOLOtiled model is based on simple adaptation of images to size and ratio of those used in the training phase of the neural network. This operation improved performance of YOLO model by about 20%, even surpassing Faster R-CNN in terms of pedestrian detection.

Detection of pedestrians using the YOLO algorithm has good accuracy, but you can improve them by simply adjusting the size of the webcam image to the size of the image used for neural network training. The split of the original high resolution image into six smaller images increased the number of detected pedestrians from 52.13% to 302.73%, as shown in Table 3. These values are proportional to the visible distance of the webcam. At short distances (All Saints’ Square), mainly pedestrians near the camera are visible in the field of view. In the case of large distances (Wawel Castle), where the nearest pedestrian is visible at a distance of 100 m, split of images into smaller tiles causes a significant change in the number of detected pedestrians. The cost of better results using the YOLOtiled method is a longer calculation time. The minimum size at which a pedestrian can be detected is approximately 15 pixels of height. So any decrease in this value due to image scaling makes it almost impossible to detect pedestrian. This is probably the main reason for the good performance of proposed new method. Reducing image resolution for large objects may result in better generalization, but for small objects the spatial features of the object may be lost. The proposed YOLOtiled method maintains the aspect ratio of the image compared to the image used during training. It can be therefore assumed that the main source of improvement in performance is the preserved size of the pedestrian.

On the other hand, in crowded scenes, the standard YOLO method works much better than YOLOtiled. This is visible when comparing Fig. 2 with Fig. 3A, or Fig. 8C with Fig. 8D and Fig. 8E.

Figure 8 The biggest difference in pedestrian detection for three webcams—less pedestrians from YOLOtiled method.

Error assigning class: (A) trashcan from Grodzka (bottom right), (B) advertisement display from Podgorze (only one real person was detected). Better detection of people in crowd from Wawel webcam (C) by YOLO compared to (D) and (E) YOLOtiled method. Photo credit: www.webcamera.pl.

Differences in the number of detected pedestrians, shown in Table 3 and Fig. 6, are often caused by errors related to incorrect classification of objects. Trashcans from Grodzka webcam (Fig. 8A) or advertisements from Podgorze Market Square (Fig. 8B) are recognized by YOLO as persons. The same problem is visible in Fig. 2, where the object detected by YOLO as people (above the car) is actually trash container. A potential problem with YOLOtiled may be double detection of a large object (e.g., bus) split into two tiles. But for pedestrians, this issue is negligible.

The practical range of pedestrian detection with YOLO can be slightly improved using the tiled method, but it is still limited to about 200 m. Beyond this distance, pedestrians are simply too small to be detected. The problem that is difficult to solve with both YOLO and YOLOtiled is the crowd. Methods based on the YOLO algorithm are not oriented to detect people in crowded scenes.

The Fig. 9 is the best illustration of changes in pedestrian activity in Cracow before and during COVID-19. A total of 5 weeks during lockdown (from the First restrictions) and a few weeks earlier show how significant was the decrease in pedestrian activity in public space. Subsequent restrictions (second and third) did not change the situation. Weekly cycles visible in almost all locations are replaced by flat lines since the first restrictions in mid-March. Until the end of the period under review, this trend remains unchanged.

Figure 9 Mean daily number of pedestrians from hourly time-lapse for four webcams in Cracow, before and during COVID-19.

Split time is set on First restrictions (13 March 2020).

During COVID-19, pedestrian activity in public spaces fell almost to zero. Observed values changed proportionally, except for two webcams. The number of detected pedestrians from the Grodzka webcam became smaller than the number of pedestrians from Podgorze Market Square (Fig. 9). This can be explained by a completely different nature of the location (urban zone). Grodzka street is occupied mainly by tourists, while Podgorze Market Square is mainly occupied by residents. This example shows the impact of COVID-19 on the tourism industry in Cracow. Reduced pedestrian mobility slows down the spread of COVID-19, but even temporary lockdown immediately affects the local community and local economy. The first demonstrations of entrepreneurs against the lockdown began in Poland on 7 May 2020.

Wellenius et al. (2020) reports that the median of changes in time spent away from from places of residence decreased by 19%. At the time of writing this article, only one quantitative assessment of mobility trends for Poland during COVID-19 was available. As reported by Aktay et al. (2020) in Google COVID-19 Community Mobility Reports for Poland (March 29, 2020), mobility trends for places like restaurants, cafes, shopping centers, theme parks and museums decreased in Lesser Poland Voivodeship (with Cracow) by 84%. This corresponds to the results from Grodzka and All Saints’ Square webcams, with a reduced number of pedestrians by 85% and 78%, respectively (Table 4). According Aktay et al. (2020) mobility trends for workplaces in Cracow decreased by 41%. This corresponds to Podgorze Market Square webcam with a 34% decrease (Table 4). The results from mobile applications developed by Google and presented in this analysis using machine learning and computer vision are very similar, despite the use of completely different methods and approaches.

The overall results of the presented analysis are strongly influenced by the location of the webcam. Two aspects are important: the urban zone, which determines the type of pedestrian (tourists or residents) and the physical location of the webcam. There are mainly tourists on Grodzka Street. On All Saints’ Square, most tourists mix with the locals. This is one of the key points within Old Town in Cracow with City Hall located nearby. Wawel Castle webcam has similar (mixed) proportions of tourists to residents as All Saints’ Square. On Podgorze Market Square tourists are rare guests. If the webcam is mounted low (All Saints’ Square), the number of correctly detected pedestrians is very high. For webcams mounted on top floors (Wawel Castle) or on roof of the building (Podgorze Market Square) chance of pedestrians detection drop significantly. Another aspect of the physical location of the webcam is distance from pedestrians. YOLO detectors are trained to detect objects on several scales, but too large a distance from the object (Wawel Castle, Podgorze Market Square) makes it impossible to identify objects, including pedestrians. For this reason, even split of HD webcam image into smaller tiles improves the accuracy of the detector, but is also limited to about 200 m.

The properties of the YOLO detector probably allow the assessment of social distance between pedestrians, which may be the next stage of data analysis. By applying a depth map and pedestrian bounding boxes, it could be possible to quantify the social distances from the webcam image. In addition, the goal-oriented tool can mask pedestrian areas, ignoring the others and thus reducing the calculation time.

Conclusions

Detection of pedestrians in urban space can be done using the YOLOv3 method and hourly time-lapse from webcams. A simple split of the HD webcam image into six smaller tiles in the proposed YOLOtiled method can increase the number of detected pedestrians by over 50%. The YOLOtiled method increases the range of pedestrian detection compared to the YOLO method, but only up to a distance estimated in this study at about 200 m. Pedestrians are not detected at longer distances. The YOLOtiled turned out to be the most efficient model compared to YOLO, Faster R-CNN and SSD.

During the COVID-19 pandemic lockdown in Cracow, from 13 March 2020 to 19 April 2020, pedestrian activity decreased by 78-85% in the tourist zone (Old Town) and by 34–55% in the residential zone. The results are very similar to the Google COVID-19 Community Mobility Reports, despite the use of various methods. Polish citizens quickly and responsibly reacted to restrictions related to the social distance, the visible manifestation of which was the limitation of pedestrian traffic in urban space during the COVID-19 pandemic.

The resulting hourly data with the number of people (pedestrians) for four webcams in Cracow from 9 June 2016 to 19 April 2020 are available for further use as CSV files.

The author would like to thank owners of the www.webcamera.pl portal for providing access to HD webcams in Cracow used in this research.

Additional Information and Declarations

Competing Interests

Author Contributions

Data Availability

The author declares that they have no competing interests.

Robert Szczepanek conceived and designed the experiments, performed the experiments, analyzed the data, prepared figures and/or tables, authored or reviewed drafts of the paper, and approved the final draft.

The following information was supplied regarding data availability:

Code and data is available at GitHub: https://gitlab.com/Cracert/pedestrian-count-covid-19.

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
