# Peer review of "Analysis of pedestrian activity before and during COVID-19 lockdown, using webcam time-lapse from Cracow and machine learning"

_PeerJ, doi:10.7717/peerj.10132_

## Round 0.1 · original submission · Major Revisions

Two reviewers with expertise relevant to this manuscript have both recommended major revisions to the paper before it can be published. They have provided some specific recommendations that include a more thorough literature review, better explanation of the tiling method and how tiling improves classification accuracy. Neither reviewer felt that the current version of the manuscript contained results that were particularly novel and I encourage you to consider ways to address this perception.

Reviewer 1 ·

Basic reporting

•The paper presents a direct application of pedestrians detection in the actual situation of the world. It compares the number of pedestrians detected before and after COVID 19 lockdown. The work is very useful and shows important findings even though the contribution is not very significant.

•The paper is easily understood nevertheless we suggest that the authors improve the english used throughout the paper and correct some typos and grammatical errors such as
-on several factors, such as, for example night lighting
- Python scripts and jupiter notebooks
- Croud counting requires development of new methods

•The literature review is not sufficient. We suggest to add a section devoted to literature review in which you can add paraghraph "People detection" and some other researches aiming at pedestrain detection using deep learning methods such as

-Jiang, Y., Tong, G., Yin, H. and Xiong, N., 2019. A pedestrian detection method based on genetic algorithm for optimize XGBoost training parameters. IEEE Access, 7, pp.118310-118321.
-Cheng, E.J., Prasad, M., Yang, J., Khanna, P., Chen, B.H., Tao, X., Young, K.Y. and Lin, C.T., 2020. A fast fused part-based model with new deep feature for pedestrian detection and security monitoring. Measurement, 151, p.107081.

-The author references the source code in too many places. It is enough to mention the source code in experimental results or in footnote in order to order to avoid too many hyperlinks.


•Please correct the way you reference tables, figures and sub figures (Fig. 8A).

Experimental design

•Please add more details explaining why the YOLOtiled method gives better results compared to YOLO.

•The author did not mention the ground thruth, regarding the true count of pedestrains provided by human observations, to assess the proposed method.

•Please add a table in which you compare YOLOtiled and YOLO in terms of computational time

Validity of the findings

•The results provided by the proposed method compared to YOLOv3 are satisfactory. But it is highly recommended to assess the performance of the proposed approach with robust scientific metrics. Moreover, comparison with state-of-the-art approaches, in the same data, will push forward the paper.

•The conclusion contains unnecessary details especially those taking about source code.

Reviewer 2 ·

Basic reporting

This research applied the YOLO machine learning algorithm for pedestrian detection and track the changes in pedestrian activity before and during the COVID-19 lockdown in Cracow, Poland.

Experimental design

1) This manuscript is just a case study of using YOLO. Each webcam image was simply divided into six smaller tiles for YOLO input. I think the authors did not improve or modify the original algorithm.

2) Please clarify why the similar dimensions/proportions between the input image and the training image can improve the accuracy of pedestrian detection. The readers expect to understand the intrinsic nature of this outcome. In Fig4a, will the accuracy be improved if the original image is cropped to remove buildings on top before dividing it into two tiles (720x720 px)?

3) The conclusion is that pedestrian activity decreased by 78-85% during the lockdown. What are the new things here? I heard similar statistics on media many times.

Validity of the findings

How was the detection accuracy of the methods validated and compared?
What is the novelty of the research? I don't see significant contributions from the manuscript.

Additional comments

- It is strange when the number of pedestrians is described in decimals, for example, 0.14 or 16.86 people. Instead of statistics in hourly, pedestrians can be counted per day for ignoring decimal usage.
- The principles of the YOLO algorithm should be introduced in more detail. A workflow should be added.

---

## Round 0.2 · accepted · Accept

Both of the original reviewers have assessed your revised manuscript and found that you had satisfied their comments and concerns. Consequently, your manuscript has been accepted.

Reviewer 1 ·

Basic reporting

The comments have been answered. The reviewers find that the responses are convincing.

Experimental design

The experimental design has been significantly improved.

Validity of the findings

The reviewers think that the comparison with different state of the art approaches, especially in the same that set, is very important. Nevertheless, we understand that in this type of applications, comparison requires time and performing materials. Hence, we suppose the comparison given in the paper is relatively enough.

Reviewer 2 ·

Basic reporting

No comment.

Experimental design

Although the suggested improvement for the original YOLO model is simple, I think this research might be helpful for a specific group of readers who are interested in pedestrian detection.

Validity of the findings

No comment.